# Assessment of Self-Reported Executive Function in Patients with Irritable Bowel Syndrome Using a Machine-Learning Framework

**DOI:** 10.3390/jcm12113771

**Published:** 2023-05-31

**Authors:** Astri J. Lundervold, Eline M. R. Hillestad, Gülen Arslan Lied, Julie Billing, Tina E. Johnsen, Elisabeth K. Steinsvik, Trygve Hausken, Birgitte Berentsen, Arvid Lundervold

**Affiliations:** 1Department of Biological and Medical Psychology, University of Bergen, Jonas Lies vei 91, 5009 Bergen, Norway; julie.billing@uib.no (J.B.); tina.johnsen@uib.no (T.E.J.); 2Department of Clinical Medicine, University of Bergen, 5021 Bergen, Norway; eline.margrete.randulff.hillestad@helse-bergen.no (E.M.R.H.); gulen.arslan.lied@helse-bergen.no (G.A.L.); elisabeth.kjelsvik.steinsvik@helse-bergen.no (E.K.S.); trygve.hausken@helse-bergen.no (T.H.); birgitte.berentsen.jacobsen@helse-bergen.no (B.B.); 3National Center for Functional Gastrointestinal Disorders, Department of Medicine, Haukeland University Hospital, 5021 Bergen, Norway; 4Department of Biomedicine, University of Bergen, 5020 Bergen, Norway; arvid.lundervold@uib.no; 5Mohn Medical Imaging and Visualization Center, Department of Radiology, Haukeland University Hospital, 5021 Bergen, Norway

**Keywords:** gut–brain axis, IBS, executive function, BRIEF-A, machine learning, feature importance

## Abstract

*Introduction*: Irritable bowel syndrome (IBS) is characterized as a disorder of the gut–brain interaction (DGBI). Here, we explored the presence of problems related to executive function (EF) in patients with IBS and tested the relative importance of cognitive features involved in EF. *Methods*: A total of 44 patients with IBS and 22 healthy controls (HCs) completed the Behavior Rating Inventory of Executive Function (BRIEF-A), used to identify nine EF features. The PyCaret 3.0 machine-learning library in Python was used to explore the data, generate a robust model to classify patients with IBS versus HCs and identify the relative importance of the EF features in this model. The robustness of the model was evaluated by training the model on a subset of data and testing it on the unseen, hold-out dataset. *Results*: The explorative analysis showed that patients with IBS reported significantly more severe EF problems than the HC group on measures of working memory function, initiation, cognitive flexibility and emotional control. Impairment at a level in need of clinical attention was found in up to 40% on some of these scales. When the nine EF features were used as input to a collection of different binary classifiers, the Extreme Gradient Boosting algorithm (XGBoost) showed superior performance. The working memory subscale was consistently selected with the strongest importance in this model, followed by planning and emotional control. The goodness of the machine-learning model was confirmed in an unseen dataset by correctly classifying 85% of the IBS patients. *Conclusions*: The results showed the presence of EF-related problems in patients with IBS, with a substantial impact of problems related to working memory function. These results suggest that EF should be part of an assessment procedure when a patient presents other symptoms of IBS and that working memory function should be considered a target when treating patients with the disorder. Further studies should include measures of EF as part of the symptom cluster characterizing patients with IBS and other DGBIs.

## 1. Introduction

The rich two-way communication between the central nervous system (CNS) and the gastrointestinal (GI) tract plays a crucial role in many aspects of human health [1]. It has become essential to understand symptom patterns in patients with functional disorders—commonly referred to as disorders of the gut–brain interaction (DGBI) [2]—as well as central nervous system disorders [3].

Irritable bowel syndrome (IBS) is the most prevalent disorder categorized within the DGBI umbrella [4]. It is a disorder defined by the Rome IV criteria as chronically recurring abdominal pain at least 1 day per week in the last three months plus two or more of the following criteria: changes associated with defecation, frequency of stool or form of stool [5]. With a high prevalence and in putting a significant burden on patients, relatives, healthcare services and society [6,7,8], it is surprising that the pathophysiology of IBS is still incompletely understood [9,10,11]. The gut–brain axis paradigm has, however, provided a valuable model to understand the close associations between pathophysiological and psychological aspects of IBS. With it, alertness is not only directed at the discomfort associated with the gastrointestinal (GI) symptoms used to define the disorder but includes symptoms associated with psychological distress and neuropsychiatric disorders [12].

A high co-occurrence of anxiety and depression in patients with IBS is well-known. This association was recently underscored in a study of a large sample of hospitalized patients with IBS, where up to 40% showed anxiety, followed by a high percentage with depression and suicidal ideation/attempts [13]. Although a diagnosis of anxiety and depression is not present in all patients with IBS, psychological distress associated with the disorder can mechanistically be explained by the bidirectional communication between the gut and the brain, including signaling via the hypothalamic–pituitary–adrenal axis (HPA-axis) [13]. Symptoms of anxiety and depression thus seem to be interwoven as part of the clinical as well as the pathophysiological picture characterizing patients with IBS [14], with  direct as well as indirect effects on GI symptoms [15] and their severity [16]. Screening for anxiety and depression should therefore always be part of a clinical assessment of patients presenting symptoms of IBS and the level and characteristics of psychological distress should be taken into account when treating patients who have the disorder [17].

Cognitive function, on the other hand, is less well-studied in patients with IBS and other DGBIs in spite of its close relationship with emotional function (see e.g., [18]) and its key role in a gut–brain perspective. Cognitive function with its neuronal substrate has been related to the dynamics of specific integrated networks in the brain [19,20,21] and also to the gut microbiome [22,23], referring to a community of trillions of microorganisms (bacteria, fungi, viruses) residing within the human gastrointestinal tract. These microorganisms are commonly referred to as “gut microbiota” and are characterized by their composition, diversity, mutualistic relationships, spatial distribution and dynamic colonization. They influence health and behavior [24], including cognitive function [23]. In a 2019 review of twelve studies on cognitive function in IBS patients, Lam et al. [25] reported impairment related to memory, attention and executive function, but the studies were characterized by several methodological shortcomings and conflicting results. Impaired performance on psychometric tests of attention and executive function was, however, confirmed by Wong et al. [26] in a study presented the same year. Executive function (EF) is an umbrella term that denotes cognitive processes of importance for problem-solving and decision tasks. EF is thus responsible for controlling and regulating most mental activities and refers to a set of higher-order cognitive processes of importance in all goal-directed behavior [27,28]. This shows that cognitive and executive functions are interconnected. EF enables us to plan, be attentive to ongoing activities and tasks and recall given instructions and is important when we need to allocate attentional resources to resolve certain emotions, thoughts and behavior.

These findings motivated the authors of the present study to investigate the presence and characteristics of EF in patients with IBS. This study is part of a larger umbrella project [29], which includes patients with IBS and healthy controls (HCs). Here, we included all participants who completed the adult version of the Behavior Rating Inventory of Executive Function (BRIEF-A). The aim was to identify the differential importance of these features when classifying a patient with IBS versus HC. As this is not known from previous studies, we selected a data analytic approach without specific predefined hypotheses. More specifically, we used a machine-learning framework in Python to explore the data, classify IBS patients versus HCs and assess feature importance. The first step was to search for a good classification model applied to data from multiple self-reported EF features. By using multiple linear and non-linear classifiers in a cross-validation scheme, we searched for a well-performing model. This model was used to identify and assess the most influential EF features. The model was trained on a subset of the data to enable testing the model in a hold-out dataset. We take into consideration that the validity of feature importance provided by a trained model is dependent on how well the model performs on unseen data. Furthermore, the procedure is highly relevant when concluding a study with a relatively small number of participants, as it would help us to evaluate the generalizability of the results.

## 2. Materials and Methods

### 2.1. Participants

The participants were part of the Bergen Brain–Gut-Microbiota (B-BGM) project at Haukeland University Hospital, Bergen, Norway, recruited through an outpatient clinic at the hospital, social media, a local newspaper and flyers. Information was provided on the project’s website (https://braingut.no (accessed on 20 April 2023)) and each potential patient was individually informed and screened before inclusion. They should all be between 18 and 65 years of age, an age range selected to make the sample comparable to previous local studies; the IBS patients should meet the Rome-IV criteria, have a score on the IBS Severity Scoring System (IBS-SSS) corresponding to moderate to severe IBS and a normal Norwegian diet the last three weeks before inclusion. Exclusion criteria included pharmacological treatment for IBS, a diagnosis of anxiety or depression, known neurological disease or neuropsychiatric disorder, treatment with antibiotics in the last three months, specific diets (including a vegetarian and vegan diet), pregnancy, previous intestinal surgery, metallic implants not compatible with MRI and having travelled outside Europe in last three weeks or planning to travel in near future. All participants gave written consent to participate. For more information about the B-BGM project, see [29].

### 2.2. Measures

#### 2.2.1. Measures Used to Describe the Participants

Age and gender were self-reported by the participants. Information about the severity of IBS symptoms and symptoms of anxiety and depression was available for a subgroup of the participants who completed the BRIEF-A questionnaire and will be presented as characteristics of the participants. IBS severity was assessed via the IBS severity scoring system (IBS-SSS), a clinical assessment tool used to determine the severity of core GI-related IBS symptoms [30]. The questionnaire consists of five items, each with a maximum score of 100 points, giving a total score of maximum 500. The severity of the IBS is defined as normal (<75), mild (75–175), moderate (175–300) and severe (>300). Most of the HCs obtained a score below 75 and the IBS symptoms in patients with IBS were predominantly defined as moderate or severe (see Figure 1b).

Symptoms of anxiety and depression were assessed via self-reports on a Norwegian version of the Hospital Anxiety and Depression Scale (HADS) [31]. The questionnaire includes 14 items, asking the participants to evaluate specific behaviors and feelings during the last week. Each question carries a maximum score of three, with a maximum score of 42. The odd-numbered items in the questionnaire measure symptoms of anxiety and even-numbered items measure symptoms of depression, providing seven answers generating an anxiety and a depression subscale, respectively, each with a maximum score of 21.

#### 2.2.2. Behavior Rating Inventory of Executive Function—Adult Version (BRIEF-A)

The Behavior Rating Inventory of Executive Function—Adult version (BRIEF-A) was used to assess core EF features. BRIEF-A is a 75-item self-reported questionnaire developed to provide adults’ (18–90 years) perspectives on their executive functions in their everyday environment [32]. Overall, BRIEF-A has shown good validity and reliability and is considered to give ecologically valid measures of EF across a wide range of medical and psychological conditions [33]. All participants completed the paper version of the questionnaire. Their responses were plotted in a commercial program providing age-corrected standardized scores (T-scores with 50 as mean and SD = 10). A T-score is defined within the clinical range if equal to or above 65. The 75 items are used to define nine clinical subscales, named inhibition, shifting, emotional control, self-monitoring, initiation, working memory, plan/organize, task monitoring and organization of materials.

### 2.3. Explorative Data Analysis and the Machine-Learning Framework

For this study, we employed the *PyCaret* 3.0 library (https://pycaret.gitbook.io (accessed on 20 April 2023)), a high-level, easy-to-use machine-learning framework in Python, to facilitate the classification of IBS patients versus healthy controls. *PyCaret*, which also includes *Numpy*, *Pandas* and *Scikit-learn*, streamlines the process of model development and evaluation by automating tasks such as data preprocessing, feature engineering and model selection. Within the *PyCaret* framework, we chose the Extreme Gradient Boosting algorithm (*XGBoost*) due to its superior performance in various classification tasks, including medical applications [34]. *XGBoost* is an advanced implementation of gradient-boosted decision trees that builds a series of weak decision tree models iteratively, combining them to generate a robust predictive model. The algorithm employs gradient boosting techniques to minimize the loss function by updating the model with gradients at each iteration, effectively reducing errors and improving classification accuracy (see, e.g., https://xgboost.ai (accessed on 20 April 2023) for more details).

In short, the analysis workflow consisted of the following steps:**Data preparation**: Importing the dataset(s), originally in *SPSS*.sav format containing IBS patients and healthy control participants with features from multiple sources such as demographic information and reports on questionnaires. This step also included data cleaning and merging using the *Pandas* data frame structure and functionality.**Explorative data analysis**: Investigating and summarizing the general characteristics of the dataset e.g., feature distributions, correlations and data visualization, employing functionality in *ydata_profiling* (https://github.com/ydataai/ydata-profiling (accessed on 20 April 2023)) and *autoviz* (https://github.com/AutoViML/AutoViz (accessed on 20 April 2023)).**Environment setup**: Initializing the *PyCaret* environment by specifying the target variable (*y*) and predictor variables and selecting the classification module. The data were randomly split (70%/30%) into a training set (X_train, y_train) and a test set (X_test, y_test). The first was used for training the model and identifying *feature importance* and the latter was used for the evaluation of classifier performance on unseen data in order to confirm *feature importance* identified in the training set and to assess generalization ability.**Model training and tuning**: Training the *XGBoost* model and comparing it with other models within the *PyCaret* framework using 10-fold cross-validation to obtain the best model.**Model evaluation**: Assessing, on the unseen test data, the classification performance of the best model (*XGBoost*), using metrics such as accuracy, sensitivity, specificity and area under the receiver operating characteristic curve (AUC).
The implementation of the complete workflow, the setup of the corresponding conda environment, the cleaned input dataset in .csv format and code for all resulting tables and figures are available as *Jupyter notebooks* at https://github.com/MMIV-ML/BRIEF-IBS (accessed on 20 April 2023).

### 2.4. Feature Importance: Permutation Importance and SHAP Values

To identify and assess the most influential features in the set of BRIEF-A predictors for the classification of IBS versus HC using *XGBoost* in the training set, we compared two quite different feature importance methods: permutation feature importance and Shapley Additive Explanations (SHAP) values.

*Permutation feature importance* is a model-agnostic method to assess the contribution of each feature to the predictive performance of the model. It measures the decrease in model accuracy when the values of a specific feature are randomly permuted, thereby disrupting the relationship between the feature and the outcome. The greater the decrease in accuracy, the higher the importance of the feature [35].

*SHAP values*, on the other hand, are based on cooperative game theory and provide a unified framework for interpreting the contribution of each feature to the prediction for a specific instance. The *SHAP value* of a feature measures its average marginal contribution to the model’s prediction across all possible feature combinations, ensuring that the contributions are fairly distributed among the features [36].

We calculated permutation feature importance using the permutation_importance function from the *scikit-learn* library (https://scikit-learn.org (accessed on 20 April 2023)). For *SHAP values*, we employed the *SHAP* library (https://github.com/slundberg/shap (accessed on 20 April 2023)) and computed TreeSHAP values specifically designed for tree-based models such as *XGBoost* ([36]). A comparison of the two methods, permutation importance and SHAP values, was performed to assess the consistency of feature importance rankings and to identify key factors that contribute to the classification of IBS patients versus healthy controls.

## 3. Results

### 3.1. Characteristics of the Participants

The BRIEF-A questionnaire was completed by 66 participants, 44 defined with IBS (77.3% females) and 22 healthy controls (HCs) (68% females). The mean age was 35 (10.2) in the IBS group and 36.1 (12.0) in the HC group. The differences between the groups were non-significant for gender and age (*p* > 0.05, Figure 1a).

In the sample with BRIEF-A scores who also have results on the HADS scores and the IBS-SSS scale, the differences in severity scores between the IBS and HC groups were all statistically significant on a Welsch’s *t*-test (*p* < 0.001), with clear separation in their distributions as measured via Cohen’s d (Table 1). The distribution of IBS-SSS scores is also illustrated in Figure 1b.

### 3.2. Distributions of EF Features in the IBS and HC Groups

The scores (according to official norms [32]) on all BRIEF-A subscales are shown in Table 2. Results from an independent sample Welsch’s *t*-test showed higher scores in the IBS than HC group on the emotional control, initiation, shifting and working memory subscales. The high effect sizes (Cohen’s d) on these measures should be noted, with the highest value for the working memory subscale.

### 3.3. Correlations and Distributions of EF Features in the IBS and HC Groups

The pairwise scatterplots in Figure 2 show the distributions of the nine BRIEF-A subscores and the least square regression lines for the IBS (in blue) and the HC group (orange) for all possible subscale pairs. Overall, the largest discrimination is found on the distribution plots for the working memory and shifting subscales and the slopes of the regression lines are similar across the two groups. In the IBS group, we find a pattern with the strongest pairwise correlations (*r* > 0.60) between self-monitoring and inhibition, emotional control and shifting, task monitoring and planning and between organization and planning and organization and task monitoring. The strongest correlations (*r* = 0.60) in the HC group were found between planning and task monitoring, added by a correlation at the same level for the pair of self-monitoring and emotional control.

### 3.4. Model Training and Tuning

Figure 3 shows the classifier performance of the training data selected when the 10-fold cross-validation procedure was employed within the *PyCaret* framework. The figure shows that the *XGBoost* algorithm was selected as the model with superior performance, with a success of classifying IBS versus HCs at an accuracy level of 81% across the folds. The *XGBoost* algorithm was therefore used to identify and assess the most influential features in the set of BRIEF-A subscales.

### 3.5. Feature Importance: Permutation Importance and SHAP Values

Figure 4 shows the feature importance identified via the following two methods: permutation feature importance and Shapley Additive Explanations (detailed description in Section 2). The two models consistently ranked the working memory subscale at the top, followed by planning, emotional control and inhibition. The importance of organization was given a lower weight when analyzed by the permutation than the *SHAP* method.

Figure 5 displays the partial dependency plots (PDP) and individual conditional expectation (ICE) plots for each of the nine BRIEF-A subscales. These plots are valuable tools to gain insights into the often non-intuitive, non-linear and complex relationship between feature values in their observable range and corresponding predicted outcome. The change in predicted outcome in each individual when the BRIEF-A subscale changes, shown by the thin lines in Figure 5, can be large within a small interval of feature values. Negative changes are noted for the self-monitor and organization subscales and a steep change is shown for emotion regulation and planning at a score between 50 and 60. The steepest changes in prediction are found for the working memory subscale, where all participants show a change even below the cutoff (i.e., 65) commonly used to alert a clinician in such cases.

#### Model Evaluation in the Test Set

The confusion matrix presented in the left panel of Figure 6 shows that 85% of the patients with IBS and 67% of the healthy controls included in the unseen test set were correctly classified. Information about the IBS-SSS scores shows that two patients with IBS with a high score are predicted as HCs and that the three HCs were falsely classified as IBS in spite of a low severity level of IBS.

The panel to the right in Figure 6 added a clinically important piece of information supporting the importance of working memory in patients with IBS. According to a cutoff score of BRIEF-A (i.e., equal to or above 65), we find that more than 40% of the patients with IBS reported a score within this clinical range on both these scales. The figure for planning was at a similarly high level, with several other scales showing that between 20 and 25% of the participants with IBS reported such high scores.

## 4. Discussion

The present study showed that patients with IBS reported significantly more severe problems than healthy controls on several BRIEF-A subscales. More than 40% of those patients reported problems on the working memory and planning subscales at a severity level that should lead to clinical attention and action, followed by high percentages for problems related to flexibility and emotional control. By using multiple classifiers in a cross-validation scheme, the *XGBoost* model was found to be the superior model. Two methods for feature importance selection consistently teased out the following most important EF features: working memory, planning and emotional control. A more detailed inspection of the partial dependency plots showed that scores on some of the BRIEF-A subscales (organization, self-monitoring and inhibition) had a negative impact in the *XGBoost* model, while the impact was strong for the working memory subscale, even at a cutoff below the one commonly used to indicate severe problems in a clinical setting (i.e., 65). Finally, the feature importance provided by the trained model was confirmed via testing the trained *XGBoost* model on the unseen hold-out dataset which correctly classified 85% of the patients with IBS.

We believe that the present study contributes by raising concerns about the presence of problems related to EF in patients with IBS. It adds information to a research field where previous studies have shown mixed results [25]. The present results are, however, similar to the ones presented by Wong et al. [26], reporting impairment on a psychometric test of EF (Wisconsin Card Sorting Test (WCST)). Previous studies have shown a weak correlation between self-reported and performance-based results [37], and discussions about their relative ecological validity are still ongoing [38]. Still, we speculate as to whether the strong working memory component of EF shown in the present study may have influenced the WCST performance presented by Wong et al. This speculation calls for further studies on working memory function in patients with IBS that include both self-reported and psychometric measures.

The strong feature importance of the working memory function in the classification model is worth further comments. The importance of working memory was already shown in the explorative part of the present study. Then, its importance was confirmed across several analytic methods. The partial dependency plot indicated that IBS could be identified even with scores below the clinical cutoff score of 65. Working memory refers to our ability to actively maintain information relevant when performing a cognitive task [39,40] and is thus crucial for problem-solving and planning in situations where we need to filter out irrelevant information. This should be of importance to patients with IBS, who experience chronically recurring abdominal pain or cramping, bloating, excess gas [41] and other related sensory and psychological discomforts that are hard to ignore.

The impact of working memory on everyday functioning is further emphasized by being a fundamental component of EF and other cognitive domains [39]. Some have also pointed to the dependencies between working memory and other aspects of EF [42]. This relational perspective on EF was partly supported by the present study. The importance of working memory problems was followed by problems related to planning and emotional regulation. Direct as well as indirect consequences of a pattern of such problems on IBS symptomatology are expected, but as far as we know these have not been studied. It is however well known that working memory can be impaired due to stress [43] and is associated with problems in patients with IBS related to fatigue [44], sleep problems [45] and not least anxiety and/or depression [46].

As pointed out in the introduction, psychological distress is common among patients with IBS and may affect and be affected by the bilateral communication between the gut and the brain. Psychological distress, commonly defined by symptoms of anxiety and depression, therefore seems to be interwoven as part of the symptomatology of the disorder [14]. Its importance as a predisposing and perpetuating factor shows that symptoms of anxiety and depression are important targets for treatment [16]. Anxiety sensitivity, an often enduring tendency to believe that symptoms are harmful to the body, is a more specific distress that has been associated with IBS [47]. Such oversensitivity and physiological anxiety have been associated with reports of problems on all BRIEF-A subscales [37]. According to Otto et al. [48], working memory function is an EF feature that is essential to health-related consequences of anxiety sensitivity: low function tends to enhance the risk both of the initiation and/or maintenance of negative health behavior. Low working memory may interfere with effortful strategies used to obtain the self-control needed to resist maladaptive and disturbing sensory, emotional and social stimuli in the everyday life of a patient with IBS. Anxiety sensitivity in patients with IBS is due to this relevant to the results of the present study, but this topic indeed deserves further study.

Although anxiety sensitivity and low working memory function may be disabling, they are also targets of several treatment programs [48]. Such programs were recently presented by the Rome working team on gut–brain behavior therapies [49]. Psychological interventions such as cognitive behavioral therapy (CBT) have been reported to be effective for treating patients with IBS [50], even with pure online administration [51]. Other studies underscore the importance of taking cognitive-emotional pre-treatment characteristics into account when deciding on a treatment option for a given patient [52]. The present study underscores the importance of assessing EF and taking the results into account when providing treatment programs such as CBT as well as those involving diet restrictions for patients with IBS. All these interventions put a strong load on cognitive functions such as planning, initiating, task monitoring, organization of materials, working memory function and emotional control—in other words, abilities related to the EF features assessed in the current study. From the results of the present study, we would thus add options that more directly target EF, such as the Goal Management Training program. It has a focus on challenges in everyday life and has successfully been included as part of training programs for psychiatric and somatic disorders (e.g., [53,54,55]). The program may help a patient with IBS to cope with competing demands or senses, such as pain and other disturbing symptoms that follow the bidirectional involvement of the gut and the brain in patients with IBS. By combining this with more traditional treatment programs, such as mindfulness, the patient may learn strategies both to regulate alertness and maintain executive control [56]. The effect may be augmented even further by adding treatments such as fecal transplantation [57]).

### Strengths and Limitations

To our knowledge, this is the first study to directly investigate reports of EF-related problems in the daily life of patients with IBS. The potential to improve awareness of these problems is a main strength of the present study. By testing a cross-validated model on an unseen dataset, the trust in the relative importance of the working memory feature of EF is strongly improved.

Several limitations should be noted. The small sample size is a major limitation. It prevents investigations of symptom patterns in subgroups of patients (e.g., those defined by stool consistency). It also hampers further investigation of specific hypotheses regarding the relationships between EF features and other characteristics of patients with IBS, such as the presence and severity of anxiety and depression, IBS severity and also gender differences. This was, however, one of the reasons for using a data-driven approach in our study. Regarding anxiety and depression, it should be noted that participants with one or both of those diagnoses were excluded from entering the study. The gender bias to the disfavor of males should also be considered as a limitation, even though this probably reflects the general predominance of females among patients with IBS. Regarding IBS symptoms, the study employed strict inclusion and exclusion criteria, leaving us with a sample where the IBS-SSS score could be used to separate the group. Future studies should therefore consider inclusion of participants along the full range of IBS-SSS scores.

The restriction of analysis to EF features may also be considered a limitation. This approach was, however, selected to identify the relative importance of different aspects of the cognitive processes involved in EF. Finally, although there are arguments for the ecological validity of self-reported EF by providing insights into the real-world function, they are also vulnerable to response biases.

## 5. Conclusions

The present study contributes by documenting EF-related problems in a large subgroup of patients with IBS and by emphasizing the importance of problems related to working memory function. Health professionals should be aware of EF-related problems and associated challenges in everyday activities and should also be aware of probable effects on adherence to treatment programs. The results should inspire further studies to include EF as one of the many features integrated into the clinical symptom pattern of IBS and other DGBIs. Longitudinal studies are definitely called for to obtain a better understanding of factors predicting healthy gut–brain communication.

## Figures and Tables

**Figure 1 jcm-12-03771-f001:**
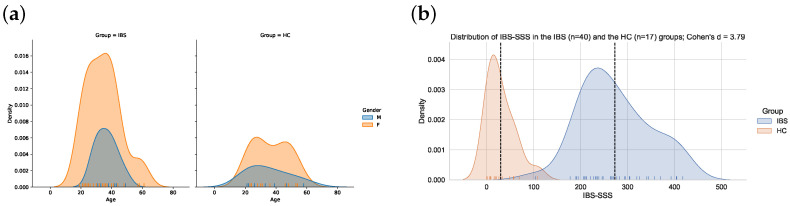
(**a**) The overlapping age distributions of males and females in the IBS (*n* = 44) and the HC (*n* = 22) group. (**b**) Distribution of IBS-SSS scores in the IBS (*n* = 40) and the HC (*n* = 17) group. The vertical dashed line denotes the mean IBS-SSS score in each group.

**Figure 2 jcm-12-03771-f002:**
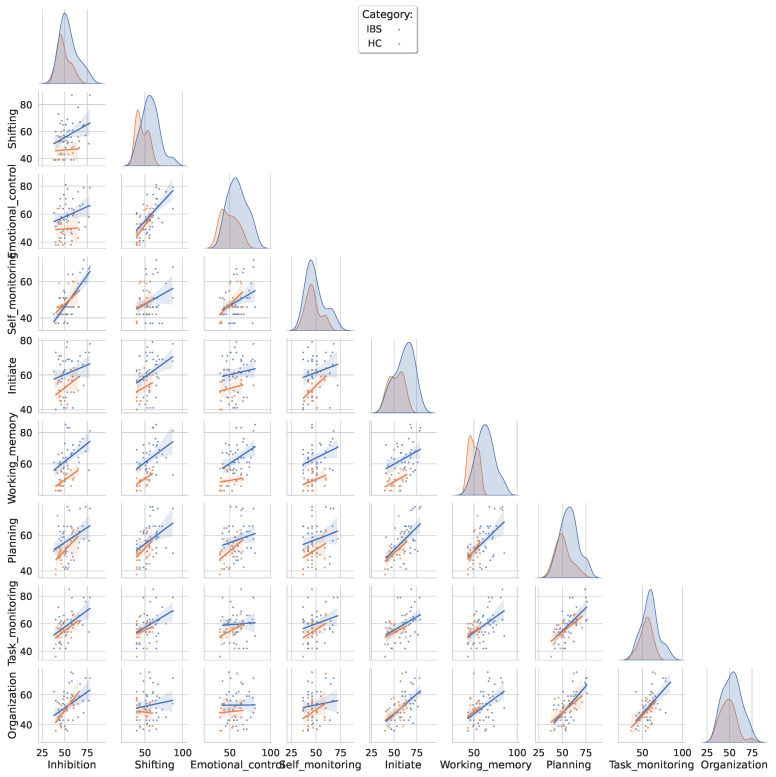
Pairwise scatterplots between all BRIEF-A subscales color-coded separately for the IBS and the HC group. The distributions are fitted with a least squares regression line with a shaded confidence interval. The diagonal entries show the group-specific kernel density estimated distributions for each BRIEF-A subscore.

**Figure 3 jcm-12-03771-f003:**
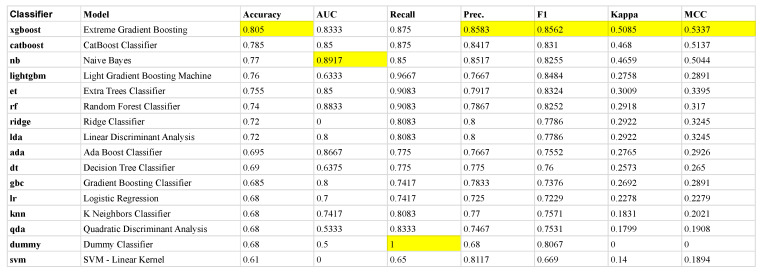
The mean metrics across each of the 10 folds in the training set. The ranking is defined with respect to mean *Accuracy* across the folds where the mean performance measures of the other metrics (AUC, Recall, Precision, F1, Kappa and MCC) for the different classifiers provided by PyCaret (left column) are given. Each classifier is applied on the same sets of folds, generated randomly from the training set, in our 10-fold cross-validation scheme. The most prominent mean values among the performance metrics are highlighted in yellow. Due to its overall best performance, we selected the *XGboost* classifier in the following analysis. Note, the *dummy* classifier makes predictions that ignore the input features, e.g., it returns the most frequent class label, serving as a baseline to compare against more complex classifiers.

**Figure 4 jcm-12-03771-f004:**
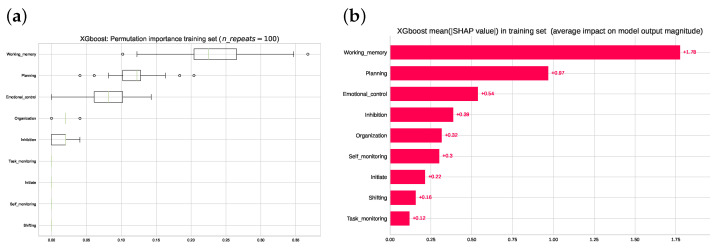
(**a**) Ranking of the permutation feature importance computed from the training set. (**b**) Ranking of mean absolute SHAP values computed in the training set. Note that the top three rankings are the same regarding order and qualitative differences for both the permutation importance method and in the SHAP values from cooperative game theory.

**Figure 5 jcm-12-03771-f005:**
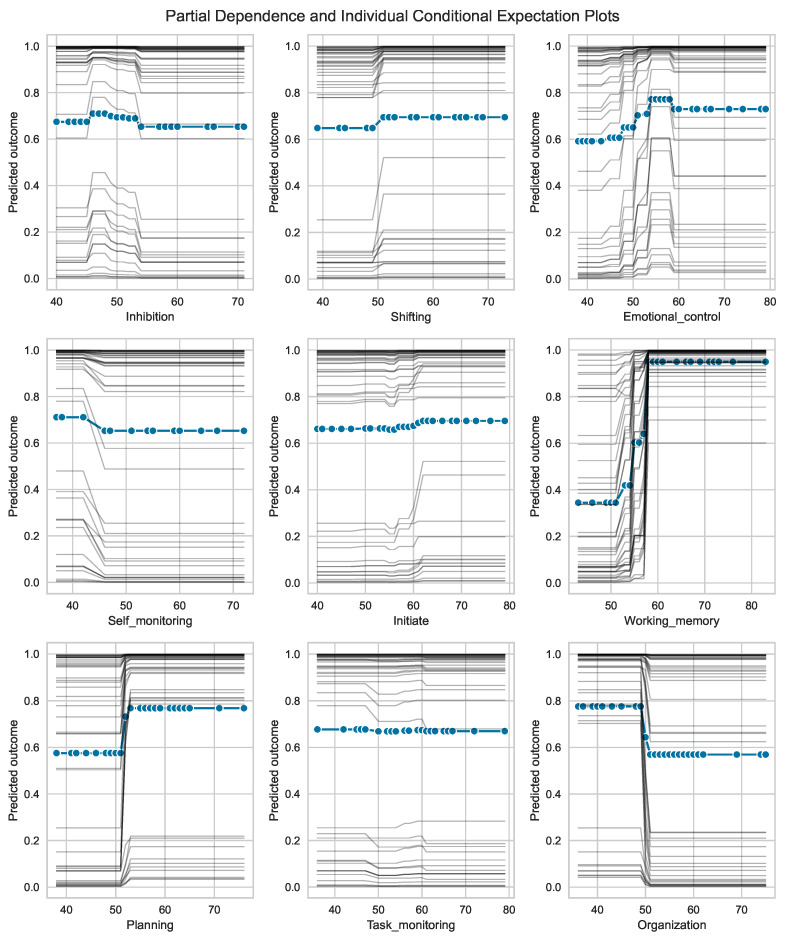
*Partial dependence plots* (PDP) and *individual conditional expectation* (ICE) plots for each of the nine BRIEF-A subscales. The collection (n=46) of thin traces across the range of BRIEF-A values, one line per subject in the training set, shows how the subject’s prediction changes when a feature (BRIEF-A subscale) changes (typically within the interval 30 to 80). Note that some of the traces are visually inseparable. The predicted outcome on the vertical axes denotes a continuous scale between 0 (=HC) and 1 (=IBS).

**Figure 6 jcm-12-03771-f006:**
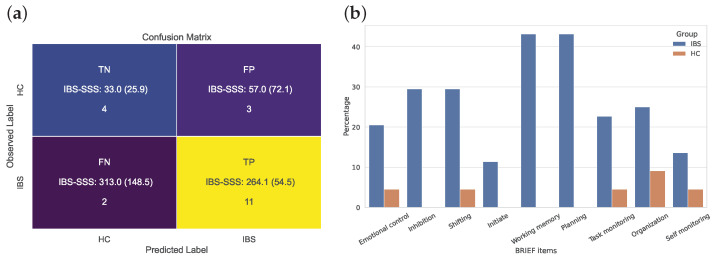
(**a**) Confusion matrix for the binary *XGBoost* classification of IBS versus HC based on the 9-dimensional feature vectors from the BRIEF domains. (**b**) BRIEF item-wise percentage of clinically impaired participants in the IBS and HC group, i.e., percentage of BRIEF variables ≥cutoff (=65) in the IBS (*n* = 44) and HC (*n* = 22) groups.

**Table 1 jcm-12-03771-t001:** BRIEF-A scores in the IBS (n=44) and HC (n=22) group.

Clinical Scale	Mean IBS/HC	SD IBS/HC	*p*-Value	Cohen’s d
Age	35.0/36.1	10.2/12.0	0.711	−0.10
Inhibition	54.5/49.0	9.8/6.9	0.011	0.62
Shifting	57.4/46.3	11.5/7.3	<0.001	1.07
Emotional_control	59.4/49.4	11.2/9.7	<0.001	0.93
Self_monitoring	49.3/47.0	9.5/7.1	0.288	0.25
Initiate	61.2/52.1	10.4/7.7	<0.001	0.95
Working_memory	63.5/49.5	10.0/5.2	<0.001	1.62
Planning	57.5/51.0	9.3/8.2	0.006	0.72
Task_monitoring	59.7/54.4	10.3/7.6	0.020	0.57
Organization	53.1/48.6	10.1/8.8	0.071	0.46

**Table 2 jcm-12-03771-t002:** Age and the HADS and IBS-SSS scores in the IBS (n=40) and HC (n=17) group.

Clinical Scale	Mean IBS/HC	SD IBS/HC	*p*-Value	Cohen’s d
HADS_anx	8.3/3.1	3.9/2.5	<0.001	1.48
HADS_dep	4.8/1.3	3.1/1.6	<0.001	1.25
HADS_tot	13.0/4.4	5.8/3.8	<0.001	1.65
IBS-SSS	273.2/29.9	73.9/29.6	<0.001	3.79

## Data Availability

The implementation of the complete workflow, the setup of the corresponding conda environment, the cleaned input dataset in .csv format and code for all resulting tables and figures are available as *Jupyter notebooks* at https://github.com/MMIV-ML/BRIEF-IBS (accessed on 20 April 2023).

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
