# Peer review of "Assessment of Self-Reported Executive Function in Patients with Irritable Bowel Syndrome Using a Machine-Learning Framework"

_jcm, 2023, doi:10.3390/jcm12113771_

Round 1
Reviewer 1 Report
This is an interesting study that explains the complexity of inflammatory bowel disease (IBD) and its unpredictable course. The main problem may be the number of subjects that were included in the study. Even though the authors considered the subject number as a limitation of their research, they should indicate if this number was selected/calculated to validate the data statistics. When describing the gut-brain axis, more information about the microbiome and its possible impact on the host's behavioral functions can be included. Most of the figures are difficult to read, the data presentation is not optimal.
Author Response
Dear reviewer!
We really appreciate your comments and have revised the manuscript accordingly. Typos and changes inspired by your comments are presented in blue text in the revised manuscript. Below you find our responses to each of your comments. Your comments are presented in red text.
This is an interesting study that explains the complexity of inflammatory bowel disease (IBD) and its unpredictable course. The main problem may be the number of subjects that were included in the study. Even though the authors considered the subject number as a limitation of their research, they should indicate if this number was selected/calculated to validate the data statistics.
Response: Our project included all participants in the Bergen-Brain-Gut study with results from the BRIEF-A questionnaire. This gave a sample size correctly described as small by the reviewer. The number of participants was restricted due to the corona pandemic, and we did not run a power analysis before concluding the study. Although we included some test statistics and p-values in our explorative analyses, the main analyses within the machine-learning framework were data-driven and run without predefined hypotheses. This is stated in the last part of the introduction of the revised manuscript. Here we have also clarified the analytic approach. Details about this approach are kept in paragraphs 2.3 and 2.4, and the source code in Python for all the analyses is ready to be openly available on GitHub if or when the manuscript is found worthy of publication. Please, tell us if you want us to make it available to you during the review of this revision.
pages 2-3, lines 78 – 93 in blue text:
“These findings motivated the present study to investigate the presence and characteristics of EF in patients with IBS. This study is part of a larger umbrella project [29]. Here we included all participants who completed the adult version of the Behavior Rating Inventory of Executive Function (BRIEF-A). The aim was to identify the differential importance of these features when classifying a patient with IBS versus HC. As this is not known from previous studies, we selected a data analytic approach without specific predefined hypotheses. The aim of the study was to identify the differential importance of these features when classifying a patient with IBS versus HC. As this is not known from previous studies, we selected a data analysis approach without specific predefined hypotheses. More specifically, we used a machine-learning framework in Python to explore the data, classify IBS patients versus HCs, and assess feature importance. The first step was to search for a good classification model applied to data from multiple self-reported EF features. By using multiple linear and non-linear classifiers in a cross-validation scheme we searched for a well-performing model. This model was used to identify and assess the most influential EF features. The model was trained on a subset of the data, to enable testing the model in a hold-out data set. By this, we take into consideration that the validity of feature importance provided by a trained model is dependent on how well the model performs on unseen data. Furthermore, the procedure is highly relevant when concluding a study with a relatively small number of participants, as it would help us to evaluate the generalizability of the results.”
When describing the gut-brain axis, more information about the microbiome and its possible impact on the host's behavioral functions can be included.
Response: Thank you for this suggestion. In the original manuscript, this is shortly described on page 2, lines 57-59, where we refer to several studies focusing on the brain and gut features of the brain-gut axis. In the revised manuscript, we have added the following sentence (lines 60-63): “These microorganisms are commonly referred to as "gut microbiota", and are characterized by their composition, diversity, mutualistic relationships, spatial distribution, dynamic colonization. By this, they influence health and behavior [24], including cognitive function [23].”
Most of the figures are difficult to read, the data presentation is not optimal.
Response: We understand that some of the figures may be challenging to read, and assume that Figures 6 and 7 were among those. Therefore, we have tried to clarify its content in the revised text. See page 6, lines 248-250 in blue text: “Figure 5 displays the partial dependency plots (PDP) and individual conditional expectation (ICE) plots for each of the nine BRIEF-A subscales. These plots are valuable tools to gain insights into the often non-intuitive, non-linear, and complex relationship between feature values in their observable range and corresponding predicted outcome.“
Table 3 and Figure 2 in the original manuscript are removed in the revised manuscript as it does not add to the information given in Figure 6 (a) and Figure 2, respectively. We assume this will make it easier to grasp the main results of the study. When it comes to data presentation, we are not quite sure about what the reviewer refers to. As mentioned above, we have incorporated a rather detailed explanation of the analytic approach in the last part of the introduction in the revised manuscript. We also assume that the details in paragraph 2.3. and 2.4 can be valuable for readers who may want to run, reproduce, or extend the analyses using the computer code that will be available upon publication. We hope that this information will clarify the presentation of the main results on feature selection and feature importance.
Description of Figure 2, page 6, lines 2020-2029: The pairwise scatterplots in Figure 2 show the distributions of the nine BRIEF-A subscores and the least square regression lines for the IBS- (in blue) and the HC group (orange) for all possible subscale pairs. Overall, the largest discrimination is found on the distribution plots for the working memory and shifting subscales, and the slopes of the regression lines are similar across the two groups. In the IBS group, we find a pattern with the strongest pairwise correlations (r > .60) between self-monitoring and inhibition, emotional control and shifting, task monitoring and planning, and between organization and planning, and organization and task monitoring. The strongest correlations (r = .60) in the HC group were found between planning and task monitoring, added by a correlation at the same level for the pair of self-monitoring and emotional control.
Reviewer 2 Report
This is a study on the brain-gut interaction and precisely the existence of executive function abnormalities in patients with IBS with 44 patients and 22 non-IBS control patients with a machine learning methodology.
I cannot assess the validity of the computer language and the machine learning methodology, I do not have the skills. There is a part of validation in this study of the computer model used which must be reread by an expert.
The conclusion of the study is that there are more significant difficulties in executive function in the IBS group than in the control group.
The results are interesting because they show a cognitive abnormality specific to the IBS population, but with one downside: either it is a new cognitive symptom documented in this population; or it is an element that stands out from the other cognitive symptoms and it becomes more interesting. The article is dense and I did not understand if anxiety and/or depression were assessed and if there was a significant difference in executive function abnormality in the depressed or anxious IBS population versus the IBS population not depressed and not anxious. If the authors have this data, it deserves to be published or explained more clearly in the study in the methods and results.
Author Response
Dear reviewer!
We really appreciate your comments and have revised the manuscript accordingly. Typos and changes inspired by your comments are presented in blue text in the revised manuscript. Below you find our responses to each of your comments. Your comments are presented in red text.
I cannot assess the validity of the computer language and the machine learning methodology, I do not have the skills. There is a part of validation in this study of the computer model used which must be reread by an expert.
Response: Thank you for being honest about this. The code in Python is didactically divided into two Jupyter notebooks, and is following standard procedures for machine learning using a high-level framework such as Pycaret, and will be openly available for inspection and use. Please let us know if you want us to share the code repository during the review of our revised manuscript. The code and the repository are developed by the last author, with a long experience in the field of computational medicine and machine learning, and is a believer in open science and reproducible research. We also hope that the inclusion of more extensive information about the analytic part at the end of the introduction would help readers who are unskilled in machine learning to better understand the results. Pages 2-3, lines 78 – 93 in blue text.
“These findings motivated the present study to investigate the presence and characteristics of EF in patients with IBS. This study is part of a larger umbrella project [29]. Here we included all participants who completed the adult version of the Behavior Rating Inventory of Executive Function (BRIEF-A). The aim was to identify the differential importance of these features when classifying a patient with IBS versus HC. As this is not known from previous studies, we selected a data analytic approach without specific predefined hypotheses. The aim of the study was to identify the differential importance of these features when classifying a patient with IBS versus HC. As this is not known from previous studies, we selected a data analysis approach without specific predefined hypotheses. More specifically, we used a machine-learning framework in Python to explore the data, classify IBS patients versus HCs, and assess feature importance. The first step was to search for a good classification model applied to data from multiple self-reported EF features. By using multiple linear and non-linear classifiers in a cross-validation scheme we searched for a well-performing model. This model was used to identify and assess the most influential EF features. The model was trained on a subset of the data, to enable testing the model in a hold-out data set. By this, we take into consideration that the validity of feature importance provided by a trained model is dependent on how well the model performs on unseen data. Furthermore, the procedure is highly relevant when concluding a study with a relatively small number of participants, as it would help us to evaluate the generalizability of the results.”
The conclusion of the study is that there are more significant difficulties in executive function in the IBS group than in the control group.
Response: Thank you for emphasizing this finding, which we consider to be a main contribution of the present study. By emphasizing this, we should convince clinicians to be aware of executive problems in patients with IBS. Impairment of EF should in other words be accounted for when selecting treatment options for individual patients with IBS and other DGBIs.
The results are interesting because they show a cognitive abnormality specific to the IBS population, but with one downside: either it is a new cognitive symptom documented in this population, or it is an element that stands out from the other cognitive symptoms and it becomes more interesting.
Response: Thank you for letting us better clarify this topic. First of all, we hope that we have conveyed the impact of executive function on a wide range of everyday situations. Although definitely not restricted to patients with IBS, a large subgroup of IBS patients show such a dysfunction at a severity level indicating a need for help. This is stated in the introduction. As the reviewer points out, there are close connections between executive function (EF) and cognitive functions like memory, attention, and more general verbal and visual aspects of intellectual function. In the revised manuscript we have included more information about these connections. First, we state that EF is an overall function involving a broad set of cognitive processes. Secondly, we assess self-reported problems, which are not always shown to be related to psychometric test results regarding EF. This is now described in more detail in the Introduction and in the Methods section on page 2. Lines 69-75: “EF is thus responsible for controlling and regulating most mental activities and refers to a set of higher-order cognitive processes of importance in all goal-directed behavior [ 27, 28 ]. This shows that cognitive and executive functions are closely interconnected. By this, EF enables us to plan, be attentive to ongoing activities and tasks, recall given instructions, and is important when we are in need to allocate attentional recourses to resolve certain emotions, thoughts, and behavior.“
The article is dense and I did not understand if anxiety and/or depression were assessed and if there was a significant difference in executive function abnormality in the depressed or anxious IBS population versus the IBS population not depressed and not anxious. If the authors have this data, it deserves to be published or explained more clearly in the study in the methods and results.
Response: We agree that information about anxiety and depression was not well integrated into the original version of the manuscript. It should have been more clearly stated that this information was included to characterize the participants. We have clarified this in the Method section of the revised manuscript. Information about anxiety and depression is now moved to the first paragraph of the revised manuscript. A more detailed analysis of subgroups with and without depression and/or anxiety is certainly of interest but is deemed to be beyond the scope of this study. Here we focus on the relative importance of nine core EF features. Furthermore, diagnoses of depression and anxiety were part of the exclusion criteria of the present study. Reasons for not including anxiety and depression in the main statistical analyses are presented in the paragraph on strengths and limitations (page 12, lines 357-369):
"Several limitations should be noted. The small sample size is a major limitation. It prevented investigations of symptom patterns in subgroups of patients (e.g., those defined by stool consistency). It also hampers further investigation of specific hypotheses regarding the relationships between EF features and other characteristics of patients with IBS, such as the presence and severity of anxiety and depression, IBS severity, and also gender differences. This was, however, one of the reasons for using a data-driven approach in our study. Regarding anxiety and depression, it should be noted that participants with one or both of those diagnoses were excluded from entering the study. The gender bias in the disfavor of males may also be considered as a limitation, even though this probably reflects the general predominance of females among patients with IBS. Regarding IBS symptoms, the study employed strict inclusion and exclusion criteria leaving us with a sample where the IBS-SSS score could be used to separate the group. Future studies should consider inclusion of participants along the full range of IBS-SSS scores. The restriction of analysis to EF features may also be considered a limitation. This approach was, however, selected to identify the relative importance of different aspects of the cognitive processes involved in EF. Finally, although there are arguments for the ecological validity of self-reported EF by providing insights into the real-world function, they are also vulnerable to response biases."
Round 2
Reviewer 2 Report
interesting corrections / adds for me